# Methylphenidate Promotes Premature Growth Plate Closure: In Vitro Evidence

**DOI:** 10.3390/ijms24044175

**Published:** 2023-02-20

**Authors:** Andrés Pazos-Pérez, María Piñeiro-Ramil, Eloi Franco-Trepat, María Guillán-Fresco, Verónica López-López, Alberto Jorge-Mora, Ana Alonso-Pérez, Rodolfo Gómez

**Affiliations:** Musculoskeletal Pathology Group, Health Research Institute of Santiago de Compostela (IDIS), Santiago University Clinical Hospital, SERGAS, 15706 Santiago de Compostela, Spain

**Keywords:** ADHD, chondrocytes, endochondral ossification, cartilage, bone, stimulants

## Abstract

It is well known that patients with attention deficit hyperactivity disorder treated with stimulants, such as methylphenidate hydrochloride (MPH), have reduced height and weight. Even though MPH has an anorexigenic effect, an additional impact of this drug on the growth plate cannot be discarded. In this study, we aimed to determine the cellular effect of MPH on an in vitro growth plate model. We tested the effects of MPH on the viability and proliferation of a prechondrogenic cell line via an MTT assay. In vitro differentiation of this cell line was performed, and cell differentiation was evaluated through the expression of cartilage- and bone-related genes as measured via RT-PCR. MPH did not alter the viability or proliferation of prechondrogenic cells. However, it reduced the expression of cartilage extracellular matrix-related genes (type II collagen and aggrecan) and increased the expression of genes involved in growth plate calcification (Runx2, type I collagen, and osteocalcin) at different phases of their differentiation process. Our results evidence that MPH upregulates genes associated with growth plate hypertrophic differentiation. This may induce premature closure of the growth plate, which would contribute to the growth retardation that has been described to be induced by this drug.

## 1. Introduction

Attention deficit hyperactivity disorder (ADHD) is a neurobehavioral disorder characterized by difficulties in maintaining attention, hyperactivity, excessive movement, and impulsiveness. ADHD patients show a persistent pattern of inattention and/or hyperactivity–impulsivity that is more frequent and severe compared to that of their peers with typical development [1,2]. The specific aetiology of this syndrome remains unknown [2], despite being the most diagnosed neuropsychiatric disorder, with a prevalence of 5–10% in children and young people and 4% in adults [3,4,5]. Untreated ADHD translates into an increased risk of accidents, mortality, depression, personality disorders, substance abuse, and school and work failure [6]. The therapy guide for this disorder specifies multidisciplinary treatment that includes parental training, behaviour therapy, and pharmacotherapy [5,6].

The most widespread treatments for ADHD are stimulants. However, there are several concerns about their use because of their side effects and the risk of abuse [5]. Among them, the most common is methylphenidate hydrochloride (MPH), which is effective for approximately 70% of patients [4,7]. MPH has strong behavioural effects, decreases hyperactivity and inattention symptoms, and causes the normalization of brain activation patterns in adulthood [7]. It increases the availability of dopamine and norepinephrine in the synaptic space [3] by inhibiting their presynaptic transporters in central adrenergic neurons [5], thus leading to stimulation of their receptors [7]. Notwithstanding, the chronic use of this drug has been described to produce a decrease in body weight gain and also growth delay in children and adolescents [8,9,10]. The mechanism by which MPH exerts this effect on growth is not known, although the most widely accepted hypothesis is that it is due to its anorexigenic effect [11,12]. However, MPH has been demonstrated to impair weight gain independently of food intake in rats [13,14]. Therefore, a deleterious effect on chondrocytes cannot be discarded [15].

Longitudinal bone growth occurs through chondrocyte proliferation and endochondral ossification in the epiphyseal growth plates [16]. In this process, growth plate mesenchymal stromal cells (MSCs) first differentiate into chondrocytes and form a cartilage template that is gradually replaced by bone [17]. During chondrocyte maturation, these cells pass through different phases: proliferative, prehypertrophic, hypertrophic, and mineralizing. Type II collagen (COL2) is expressed by chondrocytes in the proliferative phase, but its production ceases when cells enter the hypertrophic phase. Aggrecan (ACAN) expression is also increased during the proliferative phase, while type X collagen (COLX) is expressed during the prehypertrophic and hypertrophic phases [18]. Hypertrophic chondrocytes stimulate angiogenesis, osteoblasts attraction, and cartilage matrix mineralization. Furthermore, hypertrophic chondrocytes can become osteoprogenitors and osteoblasts, and directly contribute to bone formation [19]. 

The maturation of chondrocytes from the growth plate is tightly regulated [20], and any alteration to this process could lead to early closure of the growth plate [21]. Besides hormonal and nutritional factors, certain drugs can also have an impact on growth plate maturation and cause their premature closure [22]. Nonetheless, whether MPH has any effect on growth plate maturation has not been investigated yet. Moreover, few studies have evaluated the toxicity of MPH in vitro [15,23], and the only study on chondrocytes so far tested much higher concentrations of MPH than plasmatic ones [15,24]. For all these reasons, and given that in vivo studies would be influenced by MPH metabolic effects [13,14], we focused on investigating the effects of clinically relevant concentrations of MPH on chondrocyte viability and maturation in an in vitro growth plate model. Our results indicate that even though MPH did not alter chondrocyte viability, it produced changes in the expression of genes related to their differentiation process. These changes may lead to early closure of the growth plates and thus explain the association of MPH treatment with growth retardation.

## 2. Results

### 2.1. MPH Does Not Affect Chondrocyte Viability and Proliferation

In order to determine the potential effect of MPH on longitudinal bone growth, we studied its impact on chondrocytes, the only cell type present in the growth plate. Accordingly, the effects of MPH on chondrocyte viability and proliferation were analysed via MTT assay in the presence or absence of FBS.

In growth restriction conditions (culture media without FBS), no cytotoxic effect was observed on the range of MPH concentrations tested. Of note, five of these concentrations were above the plasmatic concentration described in patients treated with MPH (40 nM) [24]. The same result was obtained in proliferating chondrocytes cultured with FBS (Figure 1), which indicated that chondrocytes’ ability to proliferate was not impaired by MPH, either. 

### 2.2. MPH Treatment Alters In Vitro Chondrocyte Differentiation 

Next, considering that the specific effect of MPH on the growth plate cannot be studied in vivo due to MPH’s suppression of appetite, we conducted our research on an in vitro growth plate differentiation model. Hence, we performed differentiation of ATDC5 cells for 21 days with three different MPH doses, with one in the range of its therapeutic blood concentration (40 nM), and the others above this range (80 and 120 nM). We observed that the differentiation of ATDC5 cells was successful (Appendix A). Despite the fact that the expression of the chondrogenesis regulator SRY-box transcription factor 9 (SOX9) did not show any significant changes in our model over time (Figure 2A), the chondrocyte differentiation marker genes, the matrix protein-coding genes COL2 and ACAN (Figure 2B,C), and the hypertrophy marker gene COLX (Figure 2D) showed a significant and gradual increase in their expression from day 0 to day 21. The bone-related markers Runt-related transcription factor 2 (RUNX2), type I collagen (COL1), alkaline phosphatase (ALPL), osteoactivin (GPNMB), osteocalcin (BLGAP), and osteopontin (SPP1) exhibited a similar trend (Figure 2E–J). Other regulatory genes, such as Bone Morphogenetic Protein 2 (BMP2) and Parathyroid Hormone 1 Receptor (PTHR), did not show any significant changes due to MPH treatment (Figure 3A,B). The expression of the Wnt signaling inhibitor Dickkopf WNT Signaling Pathway Inhibitor 1 (DKK1) was not modified by MPH, either (Figure 3C). Nonetheless, Axis Inhibition Protein 2 (AXIN2), which is induced by canonical Wnt signaling, was upregulated by all doses of MPH after 7 days of differentiation (Figure 3D), while in the absence of MPH, it was upregulated only after 14 days of differentiation (Appendix A).

Interestingly, MPH treatment modulated the expression of some of these genes at different time points. After seven days of differentiation, when ATDC5 cells are supposed to be in a proliferative, not fully differentiated state, MPH treatment induced the expression of late differentiation markers such as the bone-related genes COL1 and BGLAP (Figure 2F,I). Moreover, it upregulated the expression of the osteoblastogenesis regulator RUNX2 in a dose-dependent manner (Figure 2E). Likewise, after 14 days of differentiation, the expression of RUNX2 and COL1 was upregulated by MPH treatment (Figure 2E,F). On the contrary, the gene expression levels of COL2, one of the main components of the hyaline cartilage matrix and a marker of chondrocytes in the proliferative and prehypertrophic phases, were significantly reduced (Figure 2B). The mRNA expression of another of the major constituents of the cartilage matrix, ACAN, was also inhibited after 21 days of differentiation with MPH treatment (Figure 2C). However, no changes were observed regarding the expression of COLX, ALPL, GPNMB, and SPP1 (Figure 2D,G,H,J). In addition, no significant differences were found regarding the expression of bone-related genes between MPH-treated and untreated cells after 21 days of differentiation, when ATDC5 cells are expected to already be in a hypertrophic state (Figure 2E–J).

## 3. Discussion

In this study, we used ATDC5 cells as an in vitro growth plate model [25,26] to investigate the effects of this drug on chondrocyte viability and maturation. We found that even though MPH did not modify chondrocyte viability or proliferation, it produced changes in their differentiation process. Long-term treatment with stimulants such as MPH has been associated with growth retardation in children [8,10] and with a decrease in final adult height [9]. The reason why MPH has this effect on growth is unclear, although it has been suggested to be related to stimulant-induced appetite suppression [11,12]. Nonetheless, the hypothesis that MPH can directly affect the growth plate has barely been investigated. 

Overall, the upregulation of RUNX2 and the master regulation of osteoblastogenesis, COL1, and BGLAP, together with the downregulation of COL2 and ACAN, suggests that MPH treatment provokes faster progress of the endochondral ossification process. The transcription factor RUNX2 is a critical mediator of chondrocyte hypertrophy [27] and its deficiency impairs ossification [28]. This transcription factor regulates the expression of genes related to osteoblastogenesis, including COL1 and BGLAP [29]. Type I collagen is the most abundant protein from the organic fraction of the bone extracellular matrix, which also includes non-collagenous proteins such as osteocalcin (BGLAP) [30]. Osteocalcin also acts as a promoter of osteoblastogenesis, increasing alkaline phosphatase activity and inducing its own expression, as well as that of SPP1 [31]. Upregulation of these three bone-related genes during the first days of differentiation indicates that MPH treatment triggers growth plate calcification in a premature way. 

In addition, a reduction in COL2 and ACAN expression during the last days of differentiation may also indicate premature termination of the proliferative phase of chondrocyte maturation, and thus, early initiation of the hypertrophy and mineralizing phases. The presence of type II collagen and aggrecan in the extracellular matrix is known to inhibit chondrocyte hypertrophy [32,33] and, in fact, premature hypertrophy of the growth plate chondrocytes takes place in the absence of aggrecan [34]. Despite the reduction in these two components of the cartilage extracellular matrix, the expression of SOX9 was not modified by MPH treatment. However, it is known that SOX9 is not absolutely required to activate cartilage-specific gene expression. In fact, in the absence of SOX9, SOX8 is able to perform the same functions in prechondrocytes [35]. As a further matter, AXIN2 induction by MPH indicates Wnt signaling activation. MPH has been previously described to activate Wnt transduction pathways in neural cells [36]. The activation of these signaling pathways may also contribute to alteration of the normal differentiation process [37]. In spite of the fact that other hypertrophic and osteoblastogenic markers such as COLX, ALPL, GPNMB, and SPP1 were not modulated by MPH treatment, the results obtained indicate that clinically relevant concentrations of MPH can induce premature chondrocyte hypertrophy and osteoblastogenesis and, therefore, could lead to early closure of the growth plate. 

It is known that certain drug treatments can impair the regulatory mechanisms of the growth plate and cause its premature closure [38]. Stimulant treatment may have unknown interactions with the biological factors regulating skeletal growth, including signaling proteins, transcription factors, and hormones [9,22]. These drugs, including MPH, raise the amount of dopamine in the synaptic space [3] by blocking the dopamine transporter (DAT). Interestingly, DAT^−/−^ mice have been reported to have shorter femur lengths [39], but the molecular mechanisms underlying this effect remain unknown [40].

Importantly, once the growth plate is fully closed, its function cannot be restored. For these reasons, strategies to prevent stimulant-related growth retardation should be investigated. These may include planned interruptions of medication [9] or co-treatment with other compounds. For instance, the co-administration of a retinoid antagonist has been described to prevent chondrocyte hypertrophy caused by a drug used to treat medulloblastoma, thus preserving growth plate function [22]. In the case of treatment with stimulants, their effect on appetite cannot be discarded as a factor contributing to growth delay. Nonetheless, this study demonstrates, for the first time, that MPH has a direct effect on the expression of genes related to the maturation of chondrocytes from the growth plate. 

## 4. Materials and Methods

### 4.1. Reagents

DMEM supplemented with HAM’S-F12 (DMEM/F12, ref: D8437), penicillin/streptomycin (P/S, ref: P4458), L-glutamine (ref: G7513), fetal bovine serum (FBS, ref: F7524), human apo-transferrin (ref: T4382-100MG), sodium selenite (ref: 214485-100G), TRI Reagent and RNA Isolation Reagent (ref: T9424-200ML), 3-(4,5-Dimethyl-2-thiazolyl)-2,5-diphenyl-2H-tetrazolium bromide (MTT, ref: 475989), hydrochloric acid fuming 37% (ref: 1003171011), MPH CII (ref: 1433008), and the primers listed in Table 1 were purchased from Sigma-Aldrich (St. Louis, MO, USA).

E.Z.N.A.^®^ Total RNA Kit I (ref. R6834-01) was purchased from Omega Bio-Tek (Atlanta, USA). RNase-Free DNase I (ref. D9910K) was purchased from Cientisol (A Coruña, Spain). High-Capacity cDNA Reverse Transcription Kit was purchased from Applied Biosystems Life Technologies (Waltham, MA, USA). iTaq Universal SYBR Green Supermix (ref: 1525125) was purchased from BioRad (Hercules, CA, USA). We purchased 96-well (ref: 3798) and 100 mm (ref: 430167) adherent culture plates from Corning (Phoenix, AZ, USA). Insulin (ref: 775502) was purchased from Novo Nordisk Pharma (Madrid, Spain).

### 4.2. Cell Culture

For MTT assays, ATDC5 cells were seeded in 96-well adherent culture plates (8 × 10^3^ cells per well) with a basal medium consisting of DMEM/F12, 5% FBS, 5 µg/mL transferrin, 30 nM sodium selenite, and 2% penicillin/streptomycin, and left to adhere for 6 h. After that, standard culture medium was replaced with a medium with different concentrations of MPH (5, 10, 25, 50, 75, 100, 250, and 500 nM), with and without FBS.

For cell differentiation experiments, ATDC5 cells were seeded in 100 mm adherent culture dishes (5 × 10^5^ cells per plate for day 0 and 5 × 10^4^ cells for days 7, 14, and 21). Differentiation medium, consisting of basal medium supplemented with 0.25 UI/mL insulin, was changed 3 times a week. Three concentrations of MPH were tested: 40, 80, and 120 µM.

### 4.3. MTT Assay

Four hours before the 48 h of treatment were reached, MTT compound was added to cells at a concentration of 0.5 mg/mL. After that, 100 µL of 10% SDS (Sigma-Aldrich) and 0.01% HCl solution were added per well in order to stop the enzymatic reaction, and plates were incubated overnight at 27 °C and 5% CO_2_. Finally, the next day, the plates were read using a Multiskan SkyHigh microplate spectrophotometer from Thermo Fisher (Madrid, Spain) with a wavelength of 550 nm.

### 4.4. RT-PCR

At days 7, 14, and 21, cells were lysed using TRI Reagent, and RNA was purified using E.Z.N.A. total RNA kit I, according to the manufacturer’s instructions. RNA was treated with DNase I and retro-transcribed to obtain cDNA by employing the High-Capacity cDNA Reverse Transcription Kit. Expression levels of SOX9, COL2, ACAN, COLX, RUNX2, COL1, ALPL, GPNMB, BGALP, and SPP1 were assessed via real-time PCR using iTaq Universal SYBR Green Supermix and the primers listed in Table 1. Data analysis was performed using MxPro software version 4.10, build 389 from Stratagene (San Diego, CA, USA). Relative quantification was obtained via the 2^−ΔΔCt^ method with HPRT as the reference gene. Data were normalized to a non-treated, non-stimulated control for each experiment.

### 4.5. Statistical Analysis

Data are presented as the mean ± standard error of the mean (SEM) for “n” (at least three) independent experiments. Statistical analyses were performed using GraphPad PRISM 8 software version 8.0.2, build 263 (GraphPad Software Inc., La Jolla, CA, USA). Two different types of comparison were performed: different time points vs. day 0, and each MPH-treated point vs. the untreated control from the same day. Statistically significant differences between MPH doses were determined using ANOVA followed by Fisher’s Least Significant Difference (LSD) test, grouping the samples per day of differentiation. Statistically significant differences between different time points of differentiation (without MPH) were determined by Student’s t-test. A difference was considered significant if *p*-value ≤0.05.

## Figures and Tables

**Figure 1 ijms-24-04175-f001:**
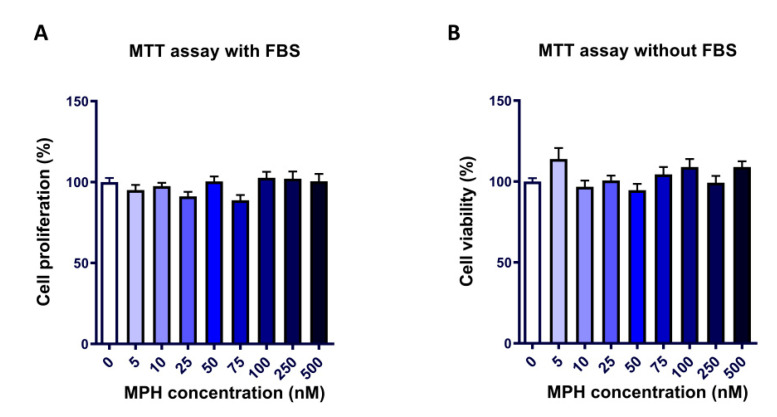
MTT assay to test the effects of 48-h treatment with MPH in ATDC5 cells on proliferation (with FBS) (**A**) and viability (without FBS) (**B**) in the culture medium (n = 3). None of the doses tested caused a significant change in the viability or proliferation of the cells. MTT: (3-(4,5-Dimethyl-2-thiazolyl)-2,5-diphenyl-2H-tetrazolium bromide); MPH: methylphenidate hydrochloride.

**Figure 2 ijms-24-04175-f002:**
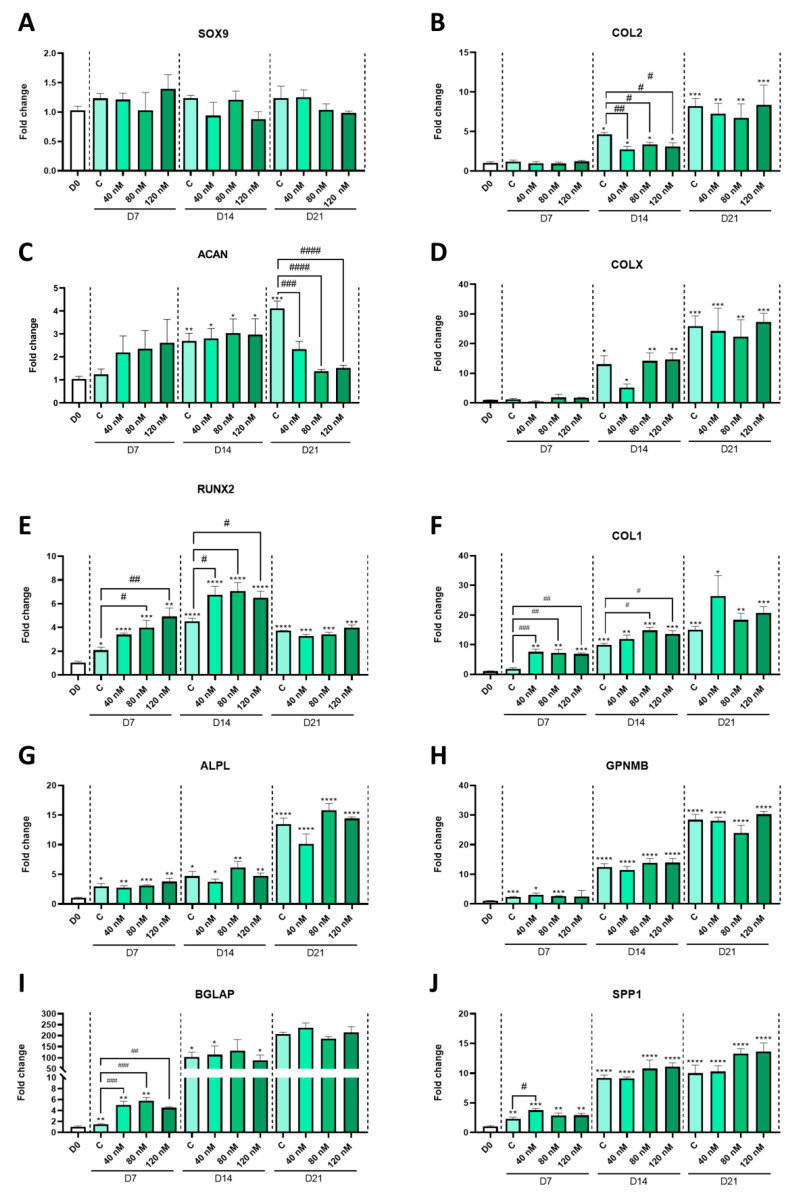
Relative expression levels of chondrocyte (**A**–**D**) and osteoblast (**E**–**J**) markers in ATDC5 cells throughout the 21 days of differentiation with or without MPH (40, 80, and 120 nM). SOX9: SRY-box transcription factor 9; COL2: type II collagen; ACAN: aggrecan; RUNX2: Runt-related transcription factor 2; COL1: type I collagen; ALPL: alkaline phosphatase; GPNMB: osteoactivin; BGLAP: osteocalcin; SPP1: osteopontin. Bars represent means with SEM (n = 3). *: statistically different vs. day 0 (*p* < 0.05); **: *p* < 0.01; ***: *p* < 0.001; ****: *p* < 0.0001; #: statistically different vs. untreated control from the same day (*p* < 0.05); ##: *p* < 0.01; ###: *p* < 0.001; ####: *p* < 0.0001.

**Figure 3 ijms-24-04175-f003:**
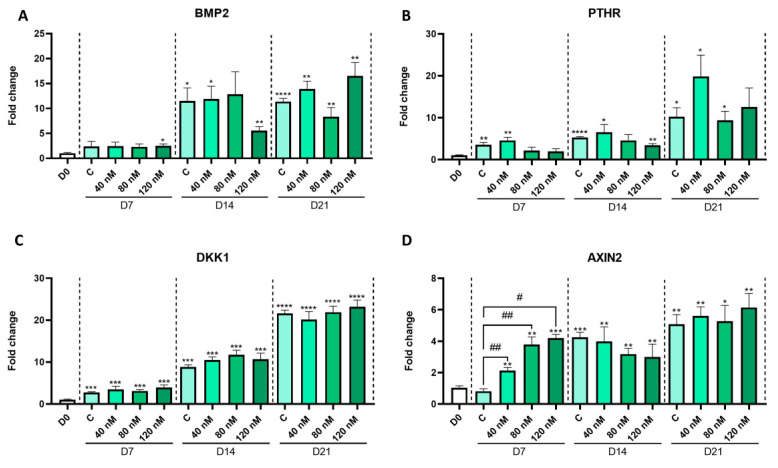
Relative expression levels of BMP2 (**A**), PTHR (**B**), and Wnt signaling-related genes (**C**,**D**) in ATDC5 cells throughout the 21 days of differentiation with or without MPH (40, 80, and 120 nM) (n = 3). BMP: Bone Morphogenetic Protein 2; PTHR: Parathyroid Hormone 1 Receptor; DKK1: Dickkopf WNT Signaling Pathway Inhibitor; AXIN2: Axis Inhibition Protein 2. Bars represent means with SEM (n = 3). *: statistically different vs. day 0 (*p* < 0.05); **: *p* < 0.01; ***: *p* < 0.001; ****: *p* < 0.0001; #: statistically different vs. untreated control from the same day (*p* < 0.05); ##: *p* < 0.01.

**Table 1 ijms-24-04175-t001:** Primers employed for real-time PCR analysis.

Gene	Symbol	Forward Primer	Reverse Primer
Hypoxanthine Phosphoribosyltransferase 1	HPRT	AGGGATTTGAATCACGTTTG	TTTACTGGCAACATCAACAG
SRY-Box Transcription Factor 9	SOX9	CTCTGGAGACTTCTGAACG	AGATGTGCGTCTGCTC
Collagen Type II	COL2	GAAGAGTGGAGACTACTGG	CAGATGTGTTTCTTCTCCTTG
Aggrecan	ACAN	CACCCCATGCAATTTGAG	AGATCATCACCACACAGTC
Collagen Type X	COLX	GCTAGTATCCTTGAACTTGG	CCTTTACTCTTTATGGTGTAGG
RUNX Family Transcription Factor 2	RUNX2	AAGCTTGATGACTCTAAACC	TCTGTAATCTGACTCTGTCC
Collagen Type I	COL1	GCTATGATGAGAAATCAACCG	TCATCTCCATTCTTTCCAGG
Alkaline Phosphatase	ALPL	TCTTCACATTTGGTGGATAC	ATGGAGACATTCTCTCGTTC
Glycoprotein Nmb	GPNMB	CAGATCAGATTCCTGTGTTTG	ACAGTATGATTGGTGGAAAC
Bone Gamma-Carboxyglutamate Protein	BGLAP	TTCTTTCCTCTTCCCCTTG	CCTCTTCTGGAGTTTATTTGG
Secreted Phosphoprotein 1	SPP1	GACCAAGGAAAACTCACTAC	CTGTTTAACTGGTATGGCAC
Bone Morphogenetic Protein 2	BMP2	CAAGAGACACCCTTTGTATG	GAATTCACAGAGTTCACCAG
Parathyroid Hormone 1 Receptor	PTHR	AGAGAAGAAGTATCTGTGGG	GATAAAGAGGATGAAGTTGAGC
Dickkopf WNT Signaling Pathway Inhibitor 1	DKK1	ATATCACACCAAAGGACAAC	CCTTCTTTAAGGACAGGTTTAC
Axis Inhibition Protein 2	AXIN2	AAGATCACAAAGAGCCAAAG	GAAAAAGTAGGTGACAACCAG

## Data Availability

The data used to support the findings of this study are contained within the article. Raw data are available from the corresponding author upon request.

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
