# Peer review of "Methylphenidate Promotes Premature Growth Plate Closure: In Vitro Evidence"

_ijms, 2023, doi:10.3390/ijms24044175_

Round 1
Reviewer 1 Report
The authors studied the impact of methylphenidate on chondrocyte differentiation in the ATDC5 cell model and demonstrated a dose-dependent increase in the level of bone-related genes over time. Chondrocyte-specific genes decreased at d14 or d21, respectively. The results suggest that methylphenidate upregulated multiple genes associated with hypertrophic differentiation of growth plate chondrocytes, which may lead to growth plate premature closure in children if exposed to methylphenidate.
Major comments
1. It would be interesting and also important to show changes of growth plate or growth plate cells in gene expression in vivo to exclude potential artefacts that may lead to misinterpretation. For in vivo studies, the control (untreated) group of animals (could be mice) may be pair fed to the MPH treated group to circumvent the reduced appetite and weight gain.
2. Have ever tried to isolate primary growth plate cells to confirm the results of MPH?
3. How many days were ATDC5 cells cultured before the MTT assay? Was there any difference in cell viability between one or two and several days?
4. How about changes in expression of other key regulatory genes, such as Wnt, PTH/PTHrP, IHH, BMP, etc.
Minor comments
1. It seems more appropriate to conclude that MPH upregulates the genes associated with growth plate chondrocyte hypertrophic differentiation.
2. Error bars for the control sample may be shown in all figures.
3. It is appropriate to run an ANOVA test for multiple comparisons of time-course curves or dose-response, which is more helpful than comparisons at each dose or time point.
4. The first sentence in the section 2.2 may be rephrased to remove repetition of words (like effect).
Author Response
Comments to Reviewer 1:
Thank you very much for your wise suggestions and for the deep review of our manuscript. We believe that your suggestions about our work have significantly improved the manuscript. As review 1 can see, we have taken into consideration all their advices. Please, find enclosed a point-to-point reply to all your queries.
Major comments:
- It would be interesting and also important to show changes of growth plate or growth plate cells in gene expression in vivo to exclude potential artefacts that may lead to misinterpretation. For in vivo studies, the control (untreated) group of animals (could be mice) may be pair fed to the MPH treated group to circumvent the reduced appetite and weight gain.
Thank you very much for this wise comment. We agree that an in vivo study would provide important information. However, the effect of MPH has been already tested in a rat model (Chirokikh et al. 2023; Uddin et al. 2018). Uddin et al. (2018) included a pair-fed control group in their study, and found that MPH weight gain was reduced in the MPH-treated group in comparison with the pair-fed untreated group, suggesting that MPH impairs weight gain independently of food intake and may produce relevant metabolic alterations. Taking this into account, we decided to perform our experiments in an in vitro model to exclude the anorexigenic effect as well as the potential metabolic effects, focusing only on the growth plate.
Chirokikh, Alexander A. et al. 2023. “Combined Methylphenidate and Fluoxetine Treatment in Adolescent Rats Significantly Impairs Weight Gain with Minimal Effects on Skeletal Development.” Bone 167.
Uddin, Sardar M.Z. et al. 2018. “Methylphenidate Regulation of Osteoclasts in a Dose- A Nd Sex-Dependent Manner Adversely Affects Skeletal Mechanical Integrity.” Scientific Reports 8(1).
- Have ever tried to isolate primary growth plate cells to confirm the results of MPH?
Thank you very much for the question. Isolating mouse growth plate chondrocytes is challenging, due to the small size of the bone. Since mouse growth plates are only a few millimetres thick, it is almost impossible to obtain a cell culture of growth plate chondrocytes without contamination with other cell types, such as bone marrow cells. This can make it difficult to accurately analyse the effect of MPH on primary growth plate chondrocyte cultures, and can lead to incorrect or misleading results.
- How many days were ATDC5 cells cultured before the MTT assay? Was there any difference in cell viability between one or two and several days?
Thank you very much for the question. ATDC5 cells were seeded in basal medium and left to adhere for 6 hours before 48-hour MPH treatment, and MTT was added to cells 4 hours before the 48 hours of treatment were reached, as stated in sections 4.2 and 4.3 of the manuscript. We have only determined cell viability at this time point (48 hours of treatment) because MTT assay is not suitable for determining viability in cultures that have reached confluence, since the proliferation of cells over time can lead to the production of high levels of formazan and thus to the saturation of the assay.
- How about changes in expression of other key regulatory genes, such as Wnt, PTH/PTHrP, IHH, BMP, etc.
Thank you for your suggestions. We have studied the expression of key genes from the suggested pathways (Wnt, PTH, BMP). Results can be found at the end of the result section and in Figure 3.
Minor comments:
- It seems more appropriate to conclude that MPH upregulates the genes associated with growth plate chondrocyte hypertrophic differentiation.
Thank you for the feedback. We have changed the conclusion to this more accurate statement in both abstract and discussion section.
- Error bars for the control sample may be shown in all figures.
Thank you for your appreciation. We have implemented the suggested changes to all the figures in the manuscript.
- It is appropriate to run an ANOVA test for multiple comparisons of time-course curves or dose-response, which is more helpful than comparisons at each dose or time point.
Thank you very much for the comment. However, the presence of a large number of conditions can make it difficult to identify any significant differences between the means of two groups, because the surrounding values may act as "background noise". In our statistical analysis, we focused on the comparisons of points with differences in only one variable (dose or time point). In this case, ANOVA test identifies differences between points in which the two variables change, and thus does not allow to determine which one (dose or time) is exerting the effect. However, we can provide all the ANOVA tests for reviewer 1 revision if required.
- The first sentence in the section 2.2 may be rephrased to remove repetition of words (like effect).
Thank you very much for your observations. We have changed the first sentence of this section of the manuscript to avoid repetition.

Reviewer 2 Report
My general evaluation for the article titled “Methylphenidate promotes premature growth plate closure: in vitro evidences” is as follows.
It is a study in the field of “The effect of MPH on the viability, proliferation, and differentiation of a prechondrogenic cell line in a growth plate in vitro model”.
It is seen that the study was organized and written in accordance with its purpose. Making the following corrections in this study will make the article stronger.
1- The abstract should be revised. The technique-method can be stated briefly. The purpose is not clearly stated.
2- In general, the English language of this article should be corrected. Professional help is recommended.
3- I think the Introduction section is sufficient.
4- I recommend that you please explain Figure 1 in detail, comparatively.
5- “4.5 Statistical analysis” is it possible for you to give more details about this topic? It is especially important that you provide detailed information about the evaluation of the data and the tests.
Author Response
Comments to Reviewer 2:
Thank you for taking the time to provide a thorough review of our manuscript and for offering valuable suggestions for improvement. We have carefully considered all of your comments and have made the necessary revisions. Please see our point-by-point response to your queries in the attached document.
Minor comments:
- The abstract should be revised. The technique-method can be stated briefly. The purpose is not clearly stated.
Thank you very much four your comment. We have implemented the suggested changes in order to clarify the methods used. Also, we have added a statement specifying the purpose of the study at the end of the Background section.
- In general, the English language of this article should be corrected. Professional help is recommended.
Thank you for your advice. This manuscript has been thoroughly revised by a professional with a C2 level of English proficiency in order to ensure the highest level of language accuracy.
- I think the Introduction section is sufficient.
Thank you for your comment.
- I recommend that you please explain Figure 1 in detail, comparatively.
Thank you very much for your comment. Following your instructions we have added more information to clarify the requested details in Figure 1 legend.
- “Statistical analysis” is it possible for you to give more details about this topic? It is especially important that you provide detailed information about the evaluation of the data and the tests.
Thank you for your observation. As explained in the statistical analysis section, we represented the data as the mean ± its standard error of all the three replicas, and then performed a t-test to determine significant changes between conditions. We have added a sentence explaining in more detail the comparisons we focused on.

Round 2
Reviewer 1 Report
1. Would recommend to include papers by Chirokikh and Uddin to explain why in vivo studies were not conducted as the challenge in running such vivo experiments is not something that cannot be overcome.
2. An ANOVA test will allow comparisons between all pairs of means. The ANOVA can be skipped if only two groups are shown for each of bar graphs, which is apparently not author’s intention.
Author Response
Thank you very much for your suggestions. Please, find enclosed a point-to-point reply to all your queries.
Comments and suggestions for authors:
1. Would recommend to include papers by Chirokikh and Uddin to explain why in vivo studies were not conducted as the challenge in running such vivo experiments is not something that cannot be overcome.
Thank you very much for this wise comment. We have added this information and both references to the introduction.
2. An ANOVA test will allow comparisons between all pairs of means. The ANOVA can be skipped if only two groups are shown for each of bar graphs, which is apparently not author’s intention.
Thank you very much for the comment. We have determined statistically significant differences between MPH doses by ANOVA followed by a Fisher Least Significant Difference (LSD) test, grouping the samples per day of differentiation, and have changed the manuscript accordingly (figures 2 and 3 and sections 2.2. and 4.5).
